# Usefulness of BRCA and ctDNA as Prostate Cancer Biomarkers: A Meta-Analysis

**DOI:** 10.3390/cancers15133452

**Published:** 2023-06-30

**Authors:** Kinga Domrazek, Karol Pawłowski, Piotr Jurka

**Affiliations:** 1Department of Small Animal Diseases and Clinic, Institute of Veterinary Medicine, Warsaw University of Life Sciences, Nowoursynowska 159c, 02-776 Warsaw, Poland; piotr_jurka@sggw.edu.pl; 2Department of Pathology and Veterinary Diagnostics, Faculty of Veterinary Medicine, Warsaw University of Life Sciences, Nowoursynowska 159c, 02-776 Warsaw, Poland; karol_pawlowski@sggw.edu.pl

**Keywords:** prostate cancer, biomarkers

## Abstract

**Simple Summary:**

Prostate cancer (PCa) represents the most common male urologic neoplasia. In the present study, the authors performed a review of scientific literature to establish the most potentially valuable biomarkers. After this a meta-analysis of two most interesting biomarkers (BRCA 1 and 2 and ctDNA) has been performed to evaluate their utility for the diagnostics, treatment, and prognosis of prostate cancer. The obtained results can help select the best diagnostic tool for early prostate diagnosis. To date, no ideal PCa biomarker has been found. Although BRCA1 and BRCA2 work well for breast and ovarian cancers, they do not seem to be reliable for prostate cancer. In our opinion, ctDNA seems to be a very promising biomarker, but still more research in this field is needed.

**Abstract:**

Prostate cancer represents the most common male urologic neoplasia. Tissue biopsies are the gold standard in oncology for diagnosing prostate cancer. We conducted a study to find the most reliable and noninvasive diagnostic tool. We performed a systematic review and meta-analysis of two biomarkers which we believe are the most interesting: BRCA (BRCA1 and 2) and ctDNA. Our systematic research yielded 248 articles. Forty-five duplicates were first excluded and, upon further examination, a further 203 articles were excluded on the basis of the inclusion and exclusion criteria, leaving 25 articles. A statistical analysis of the obtained data has been performed. With a collective calculation, BRCA1 was expressed in 2.74% of all cases from 24,212 patients examined and BRCA2 in 1.96% of cases from 20,480 patients. In a total calculation using ctDNA, it was observed that 89% of cases from 1198 patients exhibited high expression of circulating tumor DNA. To date, no ideal PCa biomarker has been found. Although BRCA1 and BRCA2 work well for breast and ovarian cancers, they do not seem to be reliable for prostate cancer. ctDNA seems to be a much better biomarker; however, there are few studies in this area. Further studies need to be performed.

## 1. Introduction

Prostate cancer represents the most common urologic neoplasia in men [1]. The incidence rate of malignancy is estimated to be 1.6 million worldwide per year [2]. The risk factors of the disease include middle age of the patient [3], obesity, family risk, and environmental influences [4]. More than 95% of prostate cancers are adenocarcinomas [5].

Prostate cancer (PCa) is most commonly diagnosed by a prostate biopsy followed by a histopathology, a rectal examination, magnetic resonance imaging, a transrectal ultrasound, and PSA (prostatic-specific antigen) detection [6,7].

The gold standard in oncology is to perform a tissue biopsy, which is an invasive method. Due to this fact, researchers are focusing on less invasive methods which allow the early diagnosis of cancer. A rapidly growing field in oncology is the study of biomarkers. The most popular and routinely performed test is evaluation of the PSA (prostate-specific antigen) level. There are some disadvantages to this biomarker. Firstly, it does not predict response to therapy [8]. Secondly, it is not specific, so there is risk of false-positive results. These facts have led to an ongoing search for the ideal biomarker for prostate cancer [9].

BRCA (BRCA1 and BRCA2) is a frequently tested biomarker in oncology. More than 3500 mutations have been reported for BRCA1 and BRCA2. It has been estimated that 0.1–0.2% of the population are carriers of those mutations [10]. The BRCA gene mutations could be inherited from either parent and passed on to one’s offspring [11]. BRCA1 is located on chromosome 11q21 and BRCA2 on chromosome 13q12–13 [12]. Both BRCA1 and BRCA2 play a role as tumor suppressor genes, which are integral for normal tissue growth control [12]. The majority of the mutations in these genes are deletions, insertions, or missense mutations, which cause premature protein terminations [12]. BRCA mutations are related mainly to breast and ovarian cancers [13], but it has been suggested that they may also affect prostate, pancreas, stomach, biliary duct, gallbladder, and colon cancers [14]. In fact, patients with BRCA mutations are exposed to a higher risk of any cancer development [10]. Due to this fact, this biomarker has been the focus of many studies of prostate cancer patients. The data in this area are contradictory. Some authors suggest that this biomarker is reliable in prostate cancer diagnosis [11,15], and others do not [16].

Another promising biomarker appears to be the circulating tumor DNA (ctDNA), which is called liquid biopsy [17]. This diagnostic tool could be used for diagnosis, expediting treatment and clinical outcomes [18]. It seems to be a valuable biomarker in many types of cancers including prostate, colorectal, lung, hepatocellular, breast, pancreatic, and gastric [18]. Liquid biopsy is highly recommended in lung and breast cancers [18]. The main advantage of ctDNA is its ability to perform the early detection of neoplasia, when the tumor DNA is circulating in the bloodstream. There are two ways that permit the translocation of DNA from the intracellular to the extracellular compartment: (I) cellular breakdown mechanisms and (II) active DNA release mechanisms [18]. It has been estimated that this biomarker can detect 69% early-stage tumors [19]. It has been found that concentration of cfDNA in blood from patients with localized cancers is in the range of 0−1000 ng/mL; in healthy individuals, the range is 0−100 ng/mL, and the range is even greater in patients with metastatic cancers [18]. However, it has been noticed that this method is not perfect. The main disadvantage is the risk of a false-negative result in the case of low ctDNA, requiring reflex testing of tumor tissue [8]. This situation could be caused by DNases or macrophages, which cause cfDNA degradation [18]. There are few extant studies of this biomarker for the diagnosis of prostate cancer.

In the present study, the authors performed a review of the available scientific literature to establish the most potentially valuable biomarkers and their utility in the diagnostics, treatment, and prognosis of prostatic cancer. The study will have a significant impact for both practitioners and other researchers, as it allows the optimal selection of biomarkers in the early diagnosis of patients with PCa and sets directions for further research. Several papers have recently been published on this topic, but none extensively evaluates the utility of ctDNA and BRCA; therefore, it was necessary to carry out this meta-analysis.

## 2. Materials and Methods

We conducted this review to examine the usefulness of selected prostate cancer biomarkers by comparing the number of PCa detected by each biomarker.

### 2.1. Study Selection

In the first stage, we conducted a systematic review of the biomarkers of prostate cancer studies. After reviewing a large number of publications, we decided to perform a meta-analysis on two biomarkers which we believe are the most interesting. The first of these was BRCA (BRCA1 and BRCA2). This is a biomarker that is a good diagnostic indicator for breast cancer in women [13]. In recent years, much research has been conducted on the usefulness of this biomarker in prostate cancer diagnosis. The second biomarker we chose was circulating tumor DNA. Although few studies have been carried out in this area, this parameter seems promising. We chose not to determine tissue and urinary biomarkers, as we believe a good biomarker is one that can be performed during a routine blood test. We followed the PRISMA statement as a checklist while preparing the manuscript. Databases such as PubMed and Google Scholar were used during the analysis. The study was performed between February and May 2023. Keywords used during the research were as follows: PCa biomarker, BRCA, BRCA1, BRCA2, BRCA prostate, BRCA1 prostate, BRCA2 prostate, ctDNA, and ctDNA prostate. The quality of the included studies has been assessed using the Newcastle-Ottawa Scale.

### 2.2. Inclusion and Exclusion Criteria

Eligible studies were included on the basis of the following inclusion criteria: (a) the types of studies included in the literature were: BRCA (BRCA1 and 2) and circulating tumor DNA (ctDNA). The exclusion criteria were as follows: (a) duplicate publications or poor-quality information (articles without raw data or with inappropriate methodology); (b) insufficient primary data, fruitless requests, or incomplete study data; (c) reviews, abstracts, commentaries, etc.; (d) infertility or fertility groups where mixed gender and data could not be separated; and (e) animal studies.

### 2.3. Data Extraction

Screening and data extraction were carried out by one researcher who first conducted literature and data screening according to the criteria developed. Another researcher ensured that the collection of the material was consistent. The established literature screening criteria were as follows: initial screening to exclude articles that clearly did not meet the criteria on the basis of the title and the abstract, followed by a detailed reading of the literature to select the final articles for inclusion in this study on the basis of the inclusion and exclusion criteria. The following data were collected from the included articles: first author, country of study, year of publication, type of sample, detection methods, and results (Figure 1).

### 2.4. Statistical Analysis

The publications that have been selected according to the predetermined criteria were statistically analyzed using GraphPad Prism 0.7 software. The data from all those publications were treated with the Chi-square test, which allowed us to determine whether the data were significantly different from what was expected and if the two categorical variables (publications, expression level) were related to each other (significance *p* value < 0.05). Then the percentage of PCa-positive patients from each selected group was calculated, and the unpaired *t*-test (significance *p* value < 0.05) was used to make a comparison between the groups (high expression circulating tumor DNA vs. low expression circulating tumor DNA, high expression of BRCA 1 or 2 vs. low/lack expression BRCA1 or 2). Also, we used the sum of cases from all selected publications to calculate the general tendency. The general tendency complied with the results from each publication calculated separately, which allowed us to see the tendency in observation.

## 3. Results

### 3.1. Characteristics of Included Studies

Our systematic search of the four databases yielded 248 articles. Forty-five duplicate articles retrieved from different databases were first excluded and, upon further examination, a further 203 articles were excluded on the basis of the inclusion and exclusion criteria established previously, leaving a total of 25 articles. There were 28,744 PCa cases in the 25 studies conducted in 9 countries: the USA, Australia, Canada, China, the United Kingdom, Israel, Japan, Korea, and Poland (Table 1). All the subjects in these 25 studies were tested or treated at hospitals. Blood, plasma, urine, tissues, buccal swabs, buffy coats, saliva, and data from a Japanese nationwide biobank or genome-sequenced samples were collected. The data extracted from each study were qualitatively synthesized and are presented in Table 1.

### 3.2. BRCA 1 and BRCA 2

We collected eighteen published studies of BRCA. Two of them were excluded due to the fact that the authors did not split the BRCA into BRCA1 and BRCA2. Sixteen studies were analyzed in which changes in the expression of the BRCA 1 and 2 were measured in patients who were PCa-positive. The percentage of cases with deregulation in the expression of BRCA was calculated (Table 2). The test for relations between variables and the studies showed that some factors of the examined patients in each experiment can play a role. However, the separate analysis to date from each study showed that that those two genes are not expressed to any great degree in PCa-positive patients. The expressions occur in a small number of PCa-positive patients. (Figure 2). As can be seen, deregulation occurs in a small percentage of cases. There was a high percentage of deregulation in studies in which the patient groups were much smaller. The larger the study group, the less frequently the deregulation of this biomarker occurred. With a collective calculation, BRCA1 was expressed in 2.74% of all cases from 24,212 patients examined and BRCA2 in 1.96% of cases from 20,480 patients. BRCA does not seem to be a promising indicator in PCa patients.

### 3.3. Circulating Tumor DNA

Seven [20,21,22,23,24,25,26] experiments were performed to measure the expression of circulating tumor DNA in the blood of 4018 PCa-positive patients. The cases were divided into groups with high expression and with low or lack of expression (Table 3).

This biomarker appears promising as an indicator of PCa. In the total calculation, we observed that 89% of the patients exhibited high expression of circulating tumor DNA. The significance was not proved by statistical comparison of those two groups (*p* = 0.25), but the results of the test are not very credible due to the small number of groups (only 7 studies); moreover, the influence of subgroups like ethnicity and age cannot be excluded. Nevertheless, the study with the highest number of examined patients (3334) showed 94% cases with high expression of circulating tumor DNA. Moreover, the studies in which 94% and 95% of the cases presented high expression examined 514 and 306 patients in total, respectively (Table 4). Five studies [20,22,23,24,26] from seven showed a similar tendency. Two studies [2,21] showed opposite results, but one presented the results of examining only eight cases. In this case, when we observed a positive correlation with the numbers of patients examined (we checked more cases in a group with high expression), it is worthwhile to continue research on this indicator (Figure 3).

## 4. Discussion

The aim of our study was to evaluate which biomarker is the most promising for prostate cancer diagnosis. After performing a systematic review of the literature published in the years 2013–2023, we subjectively decided to focus on two of the most interesting biomarkers. The reason we choose circulating tumor DNA was that this biomarker can be detected in the bloodstream and refers to DNA that comes from cancerous cells and tumors. In our opinion, the hallmark of an ideal biomarker is the ability to easily and minimally-invasively obtain material for testing. Blood is collected routinely both before surgery, during disease diagnosis, and prophylactically, so adding another parameter to the package of examinations being carried out seems an accessible solution. This biomarker is also characterized by high specificity [18]. In addition, despite the great interest of researchers in this biomarker, no meta-analysis has yet been conducted. The second biomarker we focused on was BRCA (BRCA1 and BRCA2). This indicator is the subject of a number of human cancer studies. It works well in the diagnosis of ovarian and breast cancer in women [43]. It is also the subject of a number of studies on prostate cancer in men [11,14,27,33,39,44,45,46,47]. Due to the high scientific interest in this parameter and the multitude of results available, we decided to carry out a meta-analysis in this area.

The narrative review Incorporated the most recent studies and expanded the study population. Through meta-analyses of the included studies, we successfully came to some valuable conclusions for future applications in clinical practice. The most-studied biomarkers were BRCA 1 and BRCA2, with 17 articles providing the data of their expression level in clinical prostate cancer samples. Some authors suggested that germline mutations in the BRCA genes, mainly in BRCA2, not only increase the risk of developing PCa but also have implications for the prognosis and management of the disease [46]. However, our narrative review shows a completely opposite situation. BRCA1 was a prognostic factor for 2.75% of all cases and BRCA2 for only 1.96%. This suggests that BRCA is not a promising biomarker for prostate cancer in men. On the other hand, the number and percentage of PCa-positive patients with deregulation of BRCA 1 and 2 expression varied between authors (1–51%). Depending on the methodology, the size of the study group, and the authors, the results are highly contrasted. These data suggest that this biomarker is not universal, due to the high risk of false-negative results.

In contrast to our results, the previous meta-analysis reported that being a BRCA mutation carrier (BRCA1 and/or BRCA2) is associated with a significant increase in PCa risk, with BRCA2 mutation being associated with a greater risk of PCa than BRCA1 [15]. We analyzed the possible reasons for the different results; one possible reason is that the authors estimated the risk of PCa in BRCA mutation carriers. We focused on the number of BRCA mutations in prostate cancer patients to find out if this biomarker is adequate for diagnosis of the early stage of the disease, not to calculate the risk. Our approach works better to determine which biomarker is most useful. From a clinical point of view, this is not a good approach, but the results of this narrative review may help in the future in choosing an early biomarker for prostate cancer diagnosis.

On the other hand, Fachal L. et al. also performed a meta-analysis on BRCA and did not obtain results similar to Oh et al.; nor did they show the lack of association between the BRCA1 gene and prostate cancer risk. Moreover, the results of this meta-analysis discard the involvement of BRCA1 mutations in the development of prostate cancer [16]. It is worth reflecting on the differences between these two meta-analyses and ours. It is a fact that many people have various genetic mutations that can lead to the development of cancer. It has been reported that the population frequency of pathogenic BRCA1/2 mutations is 1:400, with the exception of populations with high frequency founder mutations, such as the Ashkenazi Jewish population [48]. However, regardless of the region of the world, the results in terms of the usefulness of BRCA in the diagnosis of prostate cancer were similar. On the one hand, the fact is that not every mutation carrier will develop cancer [49]. But on the other hand, there are many various risk factors in cancerogenesis. The disease could develop without any inherited mutations. In case of prostate cancer, the most common risk factors are as follows: age (risk begins to rise after age 55), race (60% higher in Blacks than in whites), dietary factors (saturated fat, alpha-linolenic acid, red meat, dairy food), and hormonal factors (elevated intraprostatic androgens and IGF-1) [50].

Circulating tumor DNA has emerged as a minimally invasive biomarker for tumor molecular profiling and is described as liquid biopsy. Some authors suggested that ctDNA detection is rather challenging in prostate cancer [51]. In our study, high expression of circulating tumor DNA was observed in 926 of 1198 patients. Only 272 patients with prostate cancer presented low expression of this biomarker. Those results show that the detection rate is over 77%, which makes this factor promising. The ctDNA detection rate in other types of cancer is higher and can be over 90% [52], but in our opinion this PCa biomarker still compares well with others [53]. Biomarker ctDNA seems to be very promising. It has multiple potential uses in oncology such as early diagnosis, tumor molecular profiling, and early detection of resistance mutations [51]. In addition, liquid biopsy provides rapid results compared to the tissue markers. Also, importantly for patients, it is noninvasive and less expensive than traditional tissue biopsies. This method could be used for monitoring treatment progress by collecting liquid biopsies serially [17]. There are also some limitations to liquid biopsies. Some patients could obtain negative-false results due to the fact that not all patients will have detectable levels. Despite this, ctDNA testing appears to be a good prognostic marker as evidenced by numerous studies of different types of cancer [54,55,56,57].

This narrative review has some limitations. First, the literature studies were sourced from only nine countries. It should be noted that the prevalence of BRCA mutations is related to ethnicity. People with Ashkenazi Jewish heritage are at an increased risk for BRCA mutations. Dutch, French CSanadian, Icelandic, and Norwegian people may also be more likely to carry BRCA mutations. Unfortunately, data are not available for all ethnicities so we couldn’t take this into account in the meta-analysis [58]. Secondly, in the publications by BRCA, there were various diagnostic tools used (Table 1). A similar situation in the case of type of collected materials occurs (Table 1); for ctDNA, it was mainly genetic tests performed on blood or plasma (Table 1). Thirdly, our study protocol hasn’t been registered in PROSPERO.

The lack of unified materials and methods could influence the obtained results. However, even in the case of using the same method, significantly differing results have been obtained (Table 1) [2,46].

Our narrative review has some merits as well. First, we strictly followed the literature inclusion criteria. We carried out a thorough statistical analysis of the obtained results. The preparation of a meta-analysis gives a more comprehensive view, as it allows very large groups of subjects to be analyzed. Such a broad analysis would be difficult to achieve in clinical trials.

Despite some limitations, this narrative review provides some evidence that ctDNA is a much more reliable biomarker than BRCA. Given the numerous results from various authors and our meta-analysis, it seems that BRCA is not as prognostic as a PCa biomarker in the future. However, more studies in the case of ctDNA are needed. The main challenge for further studies will be standardizing methods for evaluating biomarkers.

## 5. Conclusions

To date, no ideal PCa biomarker has been found. Although BRCA1 and BRCA2 work well for breast and ovarian cancers, they do not seem to be reliable for prostate cancer. ctDNA seems to be a much better biomarker; however, there are few studies in this area and therefore we cannot tell if it is a reliable biomarker. Further studies need to be performed.

## Figures and Tables

**Figure 1 cancers-15-03452-f001:**
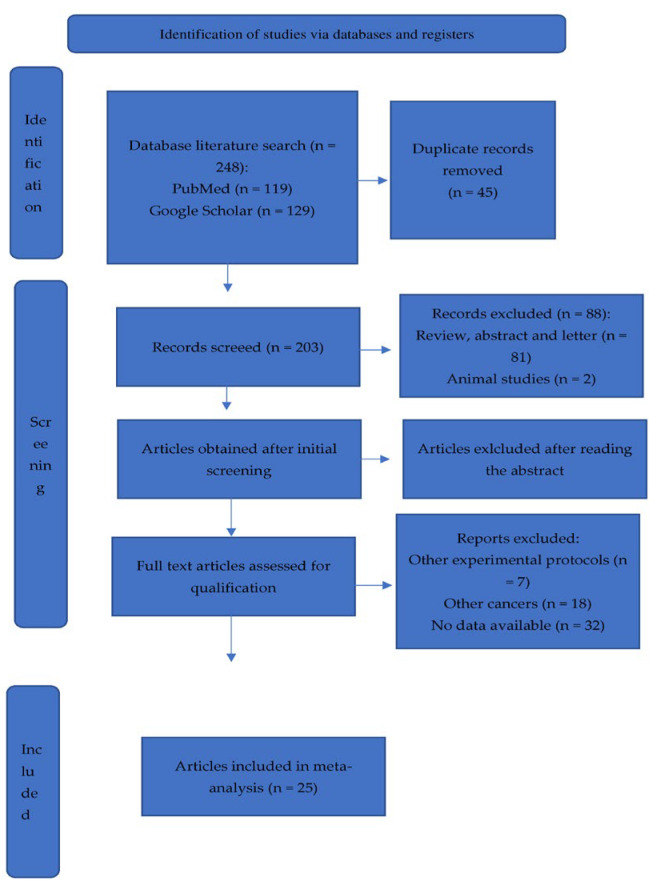
Prisma flow chart.

**Figure 2 cancers-15-03452-f002:**
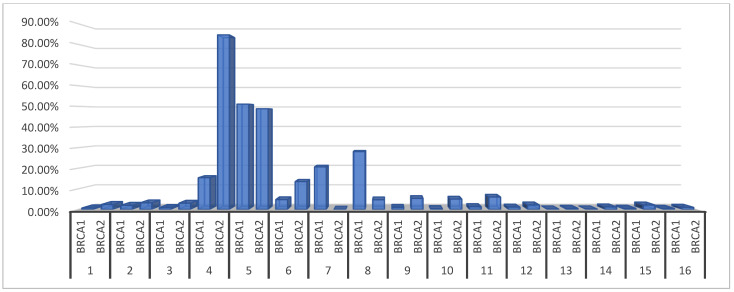
The percentage of PCa-positive patients with deregulation of BRCA 1 and 2 expression.

**Figure 3 cancers-15-03452-f003:**
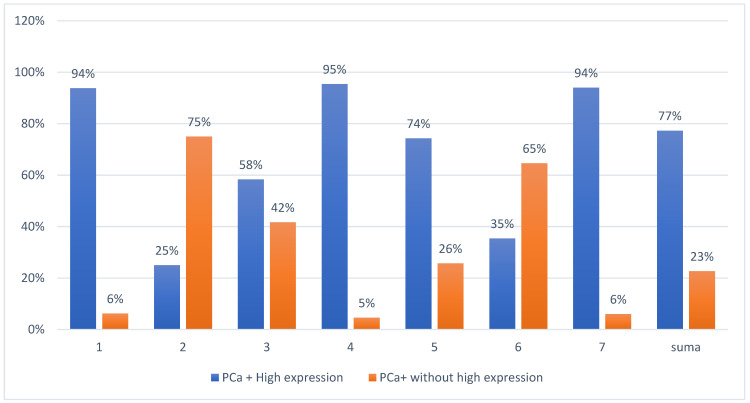
The percentage of cases of PCa-positive patients with or without high expression of circulating tumor DNA.

**Table 1 cancers-15-03452-t001:** The main characteristics of the included studies.

Biomarker	First Author	Year of Publication	Country	Method of Detection	Specimen	High Expression	Total	Citation
Circulating tumor DNA	Guru Sonpavde	2019	USA	genomic profiling	Blood	482	514	[20]
circulating tumor DNA	Edmund Lau	2020	Australia	genome sequencing	plasma	2	8	[21]
circulating tumor DNA	Sinja Taavitsainen	2019	Canada	The Guardant360 commercial ctDNA assay	blood	14	24	[22]
circulating tumor DNA	Baijun Dong	2021	China	targeted next-generation sequencing test	plasma	292	306	[23]
circulating tumor DNA	Gillian Vandekerkhove	2019	Canada	targeted next-generation sequencing test	plasma cell-free DNA (cfDNA)	26	35	[24]
circulating tumor DNA	Anuradha Jayaram	2021	UK	targeted next-generation sequencing test	plasma	110	311	[25]
circulating tumor DNA	Hanna Tukachinsky	2021	USA	Hybrid-capture-based gene panel NGS assays	plasma	3127	3334	[26]
BRCA1	David J. Gallagher	2010	USA	PCR	blood	6	832	[27]
BRCA1	Tomas Kirchhoff	2004	USA	genotyping	blood	5	251	[28]
BRCA1	Elena Castro	2013	UK	genetic testing	genome sequenced samples	18	2019	[29]
BRCA1	Mohammed Ibrahim	2018	USA	genetic testing	genome sequenced samples	2	13	[30]
BRCA1	Segal N.	2020	Israel	gene sequencing	genome sequenced samples	23	45	[31]
BRCA1	Takuhisa Nukaya	2023	Japan	PleSSision-Rapid test	formalin-fixed paraffin embedded tissues	6	126	[32]
BRCA1	Hyunho Han	2022	Korea	Targeted DNA and RNA sequencing	formalin-fixed paraffin-embedded	26	126	[33]
BRCA1	Qing Zhu	2015	China	ELISA	serum	30	107	[34]
BRCA1	Colin. C. Pritchard	2016	USA	whole-exome sequencing of germline and tumor DNA	buccal swabs, buffy coats, or whole blood	6	629	[35]
BRCA1	Matti Annala	2018	Canada	whole-exome sequencing of germline and tumor DNA	blood	1	319	[36]
BRCA1	Pedro Isaacsson Velho	2020	USA	next-generation sequencing (NGS)	saliva	2	150	[37]
BRCA1	Piper Nicolosi	2019	USA	gene sequencing	blood and saliva	38	3459	[38]
BRCA1	Yukihide Momozawa	2022	Japan	multiplex polymerase chain reaction–based target sequence method	from a Japanese nationwide biobank	14	7636	[39]
BRCA1	Yishuo Wua	2019	China	genetic testing	retrospective data	3	1694	[40]
BRCA1	Burcu F. Darst	2021	USA	gene sequencing	DNA samples	15	2770	[41]
BRCA1	Cezary Cybulski	2013	Poland	genotyping	blood	2	390	[42]
BRCA2	Qing Zhu	2015	China	ELISA	serum	5	107	[34]
BRCA2	Colin C. Pritchard	2016	USA	whole-exome sequencing of germline and tumor DNA	buccal swabs, buffy coats, or whole blood	37	692	[35]

**Table 2 cancers-15-03452-t002:** The number and percentage of PCa-positive patients with deregulation of BRCA 1 and 2 expression. Abbreviations: PCa + dysregulated = patients with prostate cancer and dysregulated BRCA biomarker; PCA + total = total number of patients with Prostate Cancer.

No	Biomarker	PCa + Dysregulated	PCa + Total	%
1 David J. Gallagher et al., 2010	BRCA1	6	832	0.7%
	BRCA2	20	832	2.4%
2 Tomas Kirchhoff et al., 2004	BRCA1	5	251	2.0%
	BRCA2	8	251	3.2%
3 Elena Castro et al., 2013	BRCA1	18	2019	8.8%
	BRCA2	61	2019	0.9%
4 Mohammed Ibrahim et al., 2018	BRCA1	2	13	3.0%
	BRCA2	11	13	15.4%
5 Segal N. et al., 2020	BRCA1	23	45	84.6%
	BRCA2	22	45	51.1%
6 Takuhisa Nukaya et al., 2023	BRCA1	6	126	48.9%
	BRCA2	17	126	4.8%
7 Hyunho Han et al., 2022	BRCA1	26	126	13.5%
	BRCA2	0	0	0%
8 Qing Zhu et al., 2015	BRCA1	30	107	28.0%
	BRCA2	5	107	4.7%
9 Colin.C. Pritchardet al 2016	BRCA1	6	629	1.0%
	BRCA2	37	692	5.3%
10 Matti Annala et al., 2018	BRCA1	1	319	0.3%
	BRCA2	16	319	5.0%
11Pedro Isaacsson Velho et al., 2020	BRCA1	2	150	1.3%
	BRCA2	9	150	6.0%
12 Piper Nicolosi et al., 2019	BRCA1	38	3459	1.1%
	BRCA2	75	3459	2.2%
13 Yukihide Momozawa et al., 2022	BRCA1	14	7636	0.2%
	BRCA2	38	7636	0.5%
14Yishuo Wua et al., 2019	BRCA1	3	1694	0.2%
	BRCA2	20	1694	1.2%
15 Burcu F. Darst et al 2021	BRCA1	15	2770	0.5%
	BRCA2	59	2770	2.1%
16 Cezary Cybulski et al., 2013	BRCA1	2	390	0.5%
	BRCA2	4	390	1.0%

**Table 3 cancers-15-03452-t003:** The number of PCa-positive patients with and without high expression of circulating tumor DNA from each publication separately, and the sum of all cases. Abbreviations: PCa+ high expression = patients with prostate cancer and high expression of this biomarker; PCa+ low/lack expression = patients with prostate cancer and low or lack expression of this biomarker; total = total number of patients with prostate cancer.

Publication	CASE		Total
	PCa + High Expression	PCa + Low/Lack Expression	
1 Guru Sonpavde et al., 2019	482	32	514
2 Edmund Lau et al., 2020	2	6	8
3 Sinja Taavitsainen et al., 2019	14	10	24
4 Baijun Dong et al., 2021	292	14	306
5 Gillian Vandekerkhove et al., 2019	26	9	35
6 Anuradha Jayaram et al., 2021	110	201	311
7 Hanna Tukachinsky et al., 2021	3127	201	3334
sum	926	272	1198

**Table 4 cancers-15-03452-t004:** The percentage of cases of PCa-positive patients with or without high expression of circulating tumor DNA. Abbreviations: PCa + high expression = patients with prostate cancer and high expression of this biomarker; PCa + low/lack expression = patients with prostate cancer and low or lack expression of this biomarker.

No	PCa + High Expression	PCa + without High Expression
1	94%	6%
2	25%	75%
3	58%	42%
4	95%	5%
5	74%	26%
6	35%	65%
7	94%	6%
sum	89%	11%

## Data Availability

The data presented in this study are available in this article.

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
