# Peer review of "Usefulness of BRCA and ctDNA as Prostate Cancer Biomarkers: A Meta-Analysis"

_cancers, 2023, doi:10.3390/cancers15133452_

Round 1

Reviewer 1 Report

A good manuscript presented by the authors. Kudos. However several concerns need to be addressed before the paper can be accepted:

1. Figure 1 quality is very bad. Kindly replace a more higher resolution figure

2. Kindly elaborate exclusion criteria of ''poor quality information"

3. Significant typo-errors were found throughout the manuscript. Kindly proofread the paper.

4. Figure 2 and table 2 are duplicated information. Kindly keep either one. 

5. Figure/table should be self-explanatory. What is 'PCa+total" and "PCa+dysregulated" means for table 2. Please define all uncommon abbreviations or representation. 

6. Figure 3 should be combined and label (A) and (B) are missing. 

7. Strongly discourage the use of black backdrop for figures.  White backdrops are more acceptable. 

8. All numbers should be carefully presented. Is it "2.75% or 2,75%" what is the meaning of 2,75% 

Significant typo-errors were found throughout the manuscript. Kindly proofread the paper.

Author Response

Dear Reviewer 1,  thank you for your suggestions. 

  1. Figure 1 has been replaced a more higher resolution
  2. Poor quality information has been elaborated
  3. Typo errors has been corrected
  4. Figure 2 has been deleted
  5. Abbreviations in the figures has been elaborated
  6. Figure 2 (3 in old version) has been corrected
  7. Backdrops has been recolored
  8. Numbers has been corrected

All changes has been marked in yellow color

Reviewer 2 Report

With pleasure, I read the paper titled: “Usefulness BRCA and ctDNA of Prostate Cancer Biomarkers: a Meta-analysis” by Domrazek and colleagues. Overall, the subject matter is of clinical interest to a wide array of readers. The topic is intellectually relevant to the journal Cancers. Collectively, the manuscript reads well and data are summarized in pertinent tables and figure, however they need revisions. The main strength of the paper includes being an updated review on the role of BRCA and ctDNA on prognosis of prostate cancer. The results are brief, but sufficient as a preliminary report. However, additional work may be needed to make a solid story. A major weakness of the study is its methodological design. I have the following comments/suggestions below.

1. Introduction. Please clearly highlight the significance of your work. It seems previous meta-analysis reports have been published previously and; therefore, it is critically central to emphasize the meaningful impact of your present work.

2. Methods. This manuscript does not seem to be a meta-analysis. It is more of a narrative review and not clearly even close to a systematic review. Please note the following comments:

a. Please mention if you followed the PRISMA statement as your checklist while preparing your manuscript.

b. Please provide the names of databases that were searched.

c. Please provide the dates of literature search from what time to what time.

d. Please provide the specific search strategy/keywords used during literature research.

e. According to Cochrane guidelines, literature search, study selection, and data collection, these steps should be completed by at least two independent coauthors, and conflicts should be resolved by consensus or consultation with a third coauthor. Your manuscript seems to mention that data collection was executed by one coauthor, hence data could be liable to mistakes.

f. Please provide information about how the quality of studies was evaluated.

g. Meta-analysis typically involves summarizing the effect sizes as odds ratio hazard ratio, event rate, etc. with 95% confidence intervals. You did not seem to provide any of that. Please see references [15] and [16].

h. A critical comment is that you need to elaborate how high versus low expression of BRCA and ctDNA was determined in these studies. Is this distinction of high versus low expression is objective and universal across studies?

i. Have you registered your study protocol in PROSPERO? Though, it is not a mandatory requirement, but it should be at least mentioned and acknowledged as a limitation if not done.

3. Results. Please consider below comments to enhance the readout of your paper.

a. Figure 1. It does not seem complete. Please check.

b. Table 2. The percentage (last column) for some do not seem correct, such as the first 2 rows. Please double check for all. Please include the reference of each study. No need for Figures 2 and 3, and their percentages could be incorporated into Table 2. Please include the citations of studies for cross-match.

c. Table 3. Please include the percentages next to the numbers and omit Table 4. Also, again, please provide the reference for each study.

4. Discussion. You referred a lot to your article as a meta-analysis, which is not true. It is more of a narrative review, and if you follow my comments in Methods, you can claim yours as a proper systematic review.  Please clearly highlight the differences between your presumed meta-analysis and the previously published meta-analysis reports (references [15] and [16] for example).

5. Conclusion. In view of the limited the evidence, the conclusion should be moderated as even ctDNA does not seem to be a promising biomarker.

6. Language. The manuscript needs, to a larger degree, extensive editing by a native English Speaker for typos.

Moderate editing of English language required.

Author Response

Dear Reviewer2,

  1. The significance of our work has been clarified in the introduction.
  2. A) yes, we followed the PRISMA statement
  3. So far it has been mentioned under the PRISMA graph. I just added also this information in the methodology.
  4. I have added this information
  5. I have added this information
  6. “Screening and data extraction was carried out by one researcher who first conducted literature screening and data, according to the criteria developed. Another researcher has been ensure that the collection of material is consistent.”
  7. The information has been added.
  8. Results obtained by athorus of cited study has been categorized by them as a high or low expression
  9. The discussion has been improved, our study has been described as a narrative review.
  10. Conclusions has been improved
  11. We planned to send this article to a language correction after final reviews.
Meta-analysis typically involves summarizing the effect sizes as odds ratio hazard ratio, event rate, etc. with 95% confidence intervals. You did not seem to provide any of that. Please see references [15] and [16].    Thank you for your valuable comment and suggestion. As it was mentioned it is more review rapport then meta-analysis especially if we consider the huge search and screening work that a meta-analysis requires.. Our aime was to focused on the preliminary study of the use of chosen markers in the diagnosis of PCa. We could not calculated RO for all for all studies because there was sometimes lack of necessary data (there was not control  healthy group). We decided to calculate odds as the frequancy of precence of marker in PCa positive patients and it was defined as the percent  to present tendencies in different observations and we focused on occurence of biomarkers only in PCa positive patients. However, it would be wort to calculate odds ratio hazard ratio, event rate, etc. on the much larger group of observation within meta-alalysis. We hope to be able to performe such analysis soon.

Round 2

Reviewer 2 Report

For most parts, the authors have addressed the comments. The manuscript can be accepted. However, extensive English editing is a must.

Extensive English editing is a must.

Author Response

Dear Reviewer,

the language has been corrected.

Thank you for your opinion. 
